# Robot Formation Performing a Collaborative Load Transport and Delivery Task by Using Lifting Electromagnets

**Celso Oliveira Barcelos** [1,†], **Leonardo Alves Fagundes-Júnior** [1,†], **Daniel Khéde Dourado Villa** [2], **Mário Sarcinelli-Filho** [2], **Amanda Piaia Silvatti** [3], **Daniel Ceferino Gandolfo** [4] and **Alexandre Santos Brandão** [1,*]

1. Robotics Specialization Center (NERo) at the Department of Electrical Engineering, Universidade Federal de Viçosa, Viçosa 36570-900, MG, Brazil
2. Department of Electrical Engineering, Federal University of Espírito Santo, Vitória 29075-910, ES, Brazil
3. Laboratory of Biomecanics Analysis (LAB) at the Department of Physical Education, Federal University of Viçosa, Viçosa 36570-900, MG, Brazil
4. Instituto de Automática, Universidad Nacional de San Juan-CONICET, Av. San Martín Oeste 1109, San Juan 5400, Argentina
* Correspondence: alexandre.brandao@ufv.br; Tel.: +55-(31)-3612-6540
† These authors contributed equally to this work.

**Featured Application: Supply delivery tasks are commonly present in hostile regions of natural disasters, in military operations, and in agricultural and urban environments. In this context, the results of this work envision the application of cargo transportation performed by a formation of robots.**

**Abstract:** This paper presents a practical validation of a heterogeneous formation of mobile robots in performing a load lifting, transportation, and delivery task. Assuming that an unmanned ground vehicle (UGV) is unable to perform a mission by itself due to the presence of an obstacle in the navigation route, an unmanned aerial vehicle (UAV) is then assigned to lift the cargo over this UGV, transport the obstacle, and deliver over another UGV. The UAV uses an electromagnetic actuator supported by a cable to pick up the load, the mass of which is 32% of that of the UAV. Experimental results demonstrate that the developed system is capable of performing cargo transport missions and can be scalable for applications such as package delivery in urban or remote areas and supply delivery in conflict or disaster zones.

**Keywords:** multirobot systems; aerial–ground cooperation; load transportation

## 1. Introduction

Over the past few years, unmanned aerial vehicles (UAVs), in particular quadrotors, have proven useful for many tasks, including multiagent missions [1–5], mapping and exploration [6–9], aerobatic performances [10], and also object manipulation for construction and transportation [11].

According to [12], in tasks involving load transportation, an unmanned aerial vehicle (UAV) has to carry a cargo by attaching it to its rigid body or suspending it by cables. In both cases, the dynamic model changes as the system becomes the UAV plus the additional cargo, and the flight controller must be able to handle this new condition. Because the rotors must provide greater thrust to lift the robot and the cargo, the energy consumption increases by attaching additional mass to the UAV. Thus, the flight time becomes shorter and shorter, which justifies homogeneous or heterogeneous cooperation in order to leverage the advantages and minimize the costs of flying each vehicle.

In practice, there are already several applications in which UAVs are used for cargo transportation, such as package delivery in urban areas [13], crop monitoring and pesticide application in precision agriculture [14], and delivery in conflict zones [15]. Thus, nowadays UAVs serve military and civilian interests.

Recent publications proposed control solutions for the manipulation and transportation of suspended payloads by using quadrotors that work cooperatively. In [12], it is stated that the control problem for the cable-suspended cargo approach is addressed in a way similar to that of a pendulum-stabilization problem. However, the cargo adds passive degrees of freedom to the UAV, directly affecting its dynamic characteristics and generating swings during flight. To do so, the controller must deal with such weight fluctuations and their consequences. The most traditional way is to stabilize and minimize cargo oscillations, neutralizing these unwanted movements. In [16], a case in which the UAV cannot lift the cargo is addressed. In this case, the strategy is to drag it laterally over the surface to the destination, whenever there is a possible route.

In contrast to most approaches, some researchers treat cable conditions as different subsystems in a hybrid dynamic model, as can be seen in [17,18], in which each subsystem has its specific and specialized controller, and a supervisory system is responsible for switching between these simpler controllers.

One way to transport a payload is by attaching it to the UAV's rigid body via mechanical or electromagnetic devices, robotic grippers, or robotic hands [19–24]. Although this option provides a simpler way by which to attach or release a payload, it reduces the agility of the rotorcraft according to [12]. Another way to perform the transport is to use suspended cables also with actuators composed of robotic arms, magnetic grippers, or mechanical devices [25–27]. This paper focuses on the latter approach, because our goal is to preserve the manoeuvrability of the UAV during execution. Furthermore, we aim to contribute to the literature with a practical demonstration of a task of grasping, lifting, carrying, and delivering a load by suspended cables. In this context, this present work aims to present an experimental implementation of a cooperative load transportation by using a heterogeneous formation. In summary, a control scheme guides a group of unmanned ground vehicles (UGVs) and a UAV and determines the best way to move a load between the UGVs. Finally, to perform the transport task, an electromagnetic system is attached to the suspended cable connected to the UAV.

To address the topics involved, this paper is divided into several sections, starting with Section 2, in which some possible real-life applications involving cooperation between aerial and ground-based multirobots are given to exemplify the importance of this work. Section 3 explains how the delivery strategy was developed based on the positioning steps and the robots. Section 5 presents some experiments performed and discusses their results, and also provides additional information about the cargo-actuator system. Finally, Section 6 highlights our concluding remarks.

## 2. Possible Practical Applications

In this section, some hypothetical situations are presented that support the importance of load transportation between aerial and ground vehicles.

Rescue operations present numerous situations in which it is necessary to apply techniques related to cargo transport. In the first example, illustrated in Figure 1, let's assume that an ambulance is going to help a victim of a traffic accident in a very congested avenue or a hard-to-cross bridge. In such situation, a UAV can assist in these emergencies by transporting first aid equipment or even specific/appropriate medical devices between the nearest hospital to the accident site where the rescue service is being performed.

To further complement some rescue operations, Figure 2 presents a didactic illustration of a situation in which a rescue team is trying to contain a fire in a burning building. In this hypothetical situation, there are civilians present inside the structure, and there is no easy access to the interior of the building due to the fire and the imminent risk of the structure collapsing. Once the rescue team is unable to access the interior of the building, immediate relief equipment can be sent to victims, such as a gas mask or water for example, aiming to increase the chances of rescuing a more significant number of victims still alive.

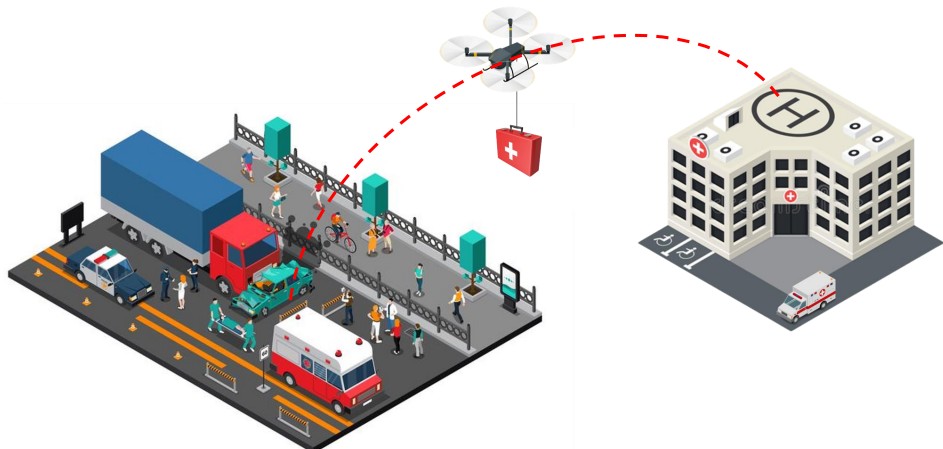

**Figure 1.** Example of situation of emergency delivery.

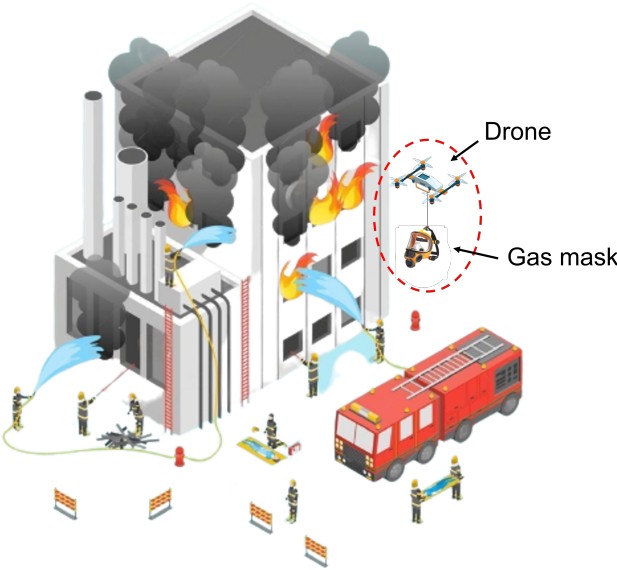

**Figure 2.** Example of situation of rescue operation.

*Delivery Operations*

Drone deliveries are one of the main trends for the future of e-commerce. In fact, with the expansive growth of the industry in recent years, companies in the sector are looking for increasingly advanced solutions to stand out in this extremely competitive market. Given this, the delivery of products by using drones is a very promising strategy for the future [28].

As another practical example in a real-world scenario, imagine that a ground delivery vehicle is in transit to deliver a large number of goods. Suppose now that this vehicle is crossing a long-span bridge, at the site of a mechanical problem, congestion due to an accident, or even civic maintenance on the bridge structure. In this situation, this vehicle will be stopped on the bridge until traffic is cleared, which may seem normal as this happens with high frequency in today's urban environments. However, if we suppose that vehicle is transporting hospital supplies, such as vaccines or even human organs, we have a situation in which every second of delay could risk lives. Thinking of such situations, it is proposed that a UAV goes to the ground vehicle that is stopped in traffic. Then, it flies carrying the cargo to a desired coordinate, where there will be a third vehicle waiting for it. Figure 3 illustrates such a situation.

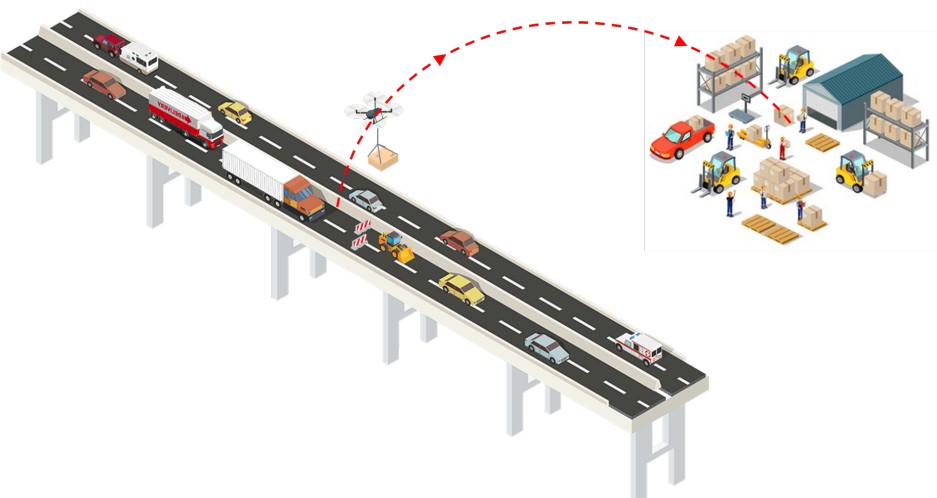

**Figure 3.** Example of delivery operation.

After presenting some hypothetical applications related to suspended load transportation with a heterogeneous robot team, we present in the sequel our a cargo transportation strategy in cases in which a UGV cannot fulfill a delivery task, and a UAV comes to help it in the accomplishment of the mission.

## 3. The Cooperative Cargo Transportation Strategy

Among some applications listed, this manuscript focuses its efforts on implementing a control strategy for transport with a heterogeneous formation during lifting, transposing, and delivering loads. To stress the multirobot collaboration, we assume the environment has a physical barrier that makes it impossible for ground robots to cross from one side to another, for instance, a UGV trying to reach the opposite side of a broken bridge. Figure 4a illustrates the ground robots P1 and P2, the aerial vehicle B1, the cargo C1, and also a trailer T1 used to transport B1.

Because there is an obstacle positioned between the UGVs, we assume that it will be necessary to use three robots, two UGVs (P1 and P2, respectively) and one UAV (B1) for practical validation of a real cargo transport strategy. Once the robot P1 cannot overcome the imposed barrier, P2 reaches the other side of the barrier at a predefined coordinate, and then receives the load, which is lifted, transported, and delivered by a UAV.

Our delivery strategy is composed by four steps, as follows.

**1:**    Cargo (C1) needs to be taken from city A to city B, but we previously know of the existence of a recently collapsed bridge between the two cities. A single vehicle (P1) cannot reach city B, and requires a second vehicle (P2) on the opposite side to accomplish the delivery task.

**2:**    Once P1 arrives at the bridge, P2 is called to pick up the load on the other side of the bridge.

**3:**    After P2 arrives to serve P1, the load must be transferred between them by using B1. With P1 and P2 waiting at rest, B1 begins the aerial transposition above the fallen bridge by picking up the load from P1 and placing it over P2. Once this is done, B1 goes back to meet P1 and lands on it.

**4:**    After overcoming the obstacle, the cargo over P2 goes to the desired destination, whereas P1 and B1 returns to city A to receive a new delivery order.

Figure 4 presents in detail the events of the delivery task described above, and how it will be implemented in practice. Initially, Figure 4a shows the agents that compose the entire formation and how they are arranged in space.

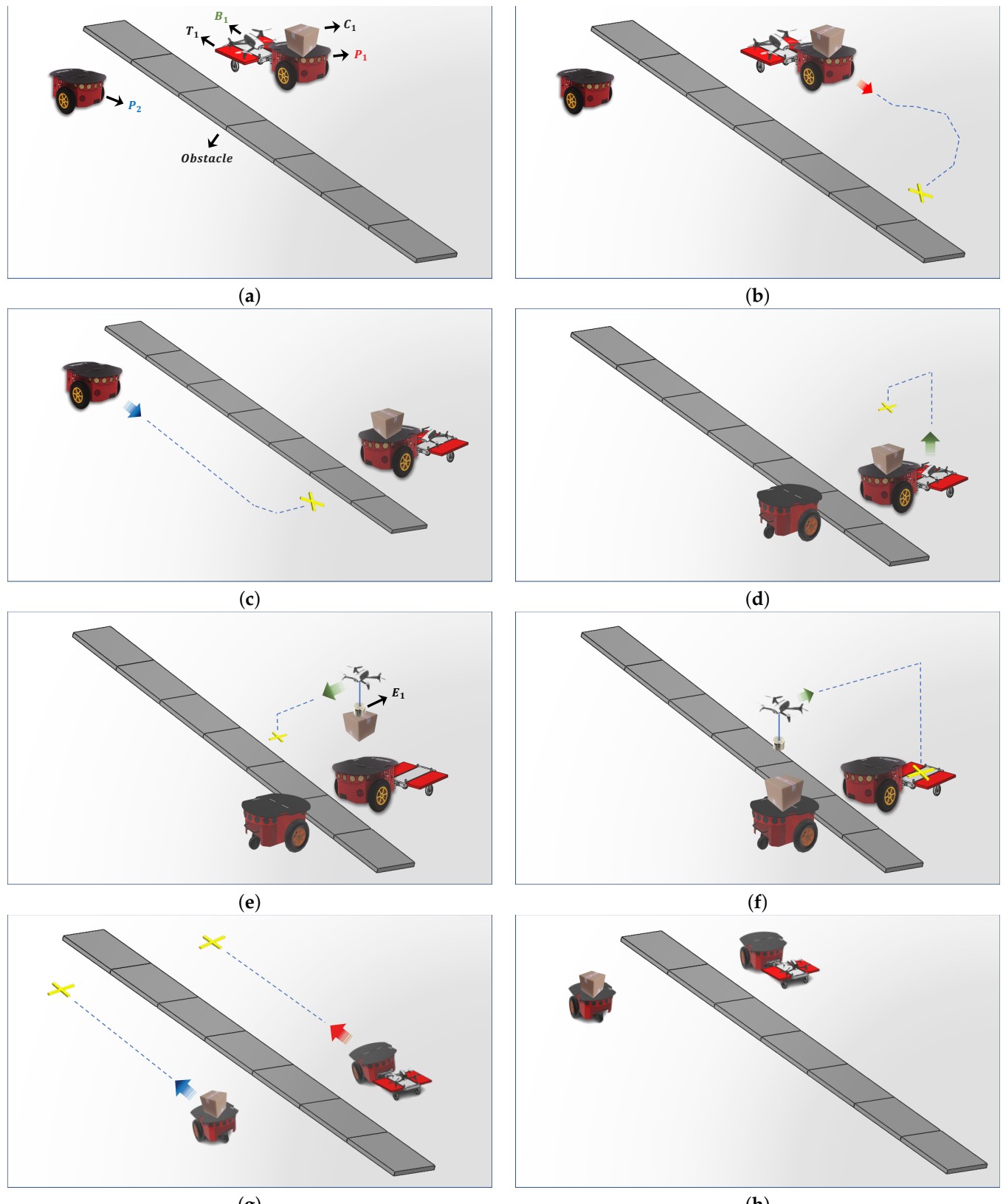

**Figure 4.** Sequence of events of formation during the experiment. (**a**) Agents of heterogeneous formation. (**b**) B1 starts moving towards the desired point. (**c**) P2 starts its movement to give assistance to P1. (**d**) B1 takes off to pick up C1. (**e**) B1 passes with C1 under the obstacle. (**f**) B1 delivers C1 on P2 and returns to T1 to land. (**g**) All formation returns to their respective destination. (**h**) Task is completed successfully.

Figure 4b, P1 goes to the desired target close to the obstacle. At the same time, B1 is carried on the trailer T1; note also that C1 is carried on P1. Figure 4c shows P2 starting this movement; it goes to the opposite side of the obstacle to receive C1. Then, B1 takes off to position itself over P1. After that, B1 captures C1 by using the electromagnet E1 (see Figure 4d). Now, B1 transports C1 from P1 to P2, as depicted in Figure 4e. Being over P2, B1 decouples C1 and returns to land on T1 after stabilizing the pendulum motion of E1 (see Figure 4f). In the sequel, P1 and P2 go to their respective pickup/delivery points (Figure 4g). Finally, the transportation task is successfully completed as shown in Figure 4h.

It is important to stress that P1 cannot move freely during the positioning task, because of T1 attached to it. In other words, there are some maneuvers that are impossible to do, such as sharp bends, where T1 can collide with P1. To overcome this movement constraint, we define a set of way-point that results in a smooth and approaching maneuver. Such a strategy is presented with clarity in Figure 5. Therefore, to pose P1 perpendicular to the obstacle, its initial movement consisted of a straight segment until the middle of the path and, from then on, a curve modeled by using small, straight segments. In turn, the path of P2 is simpler, once it is enough to inform the desired point next to P1, and then a point close to the obstacle to make it rotate 90º and be positioned facing P1. Finally, after performing the load transposition, P1 and P2 to return to their respective home points.

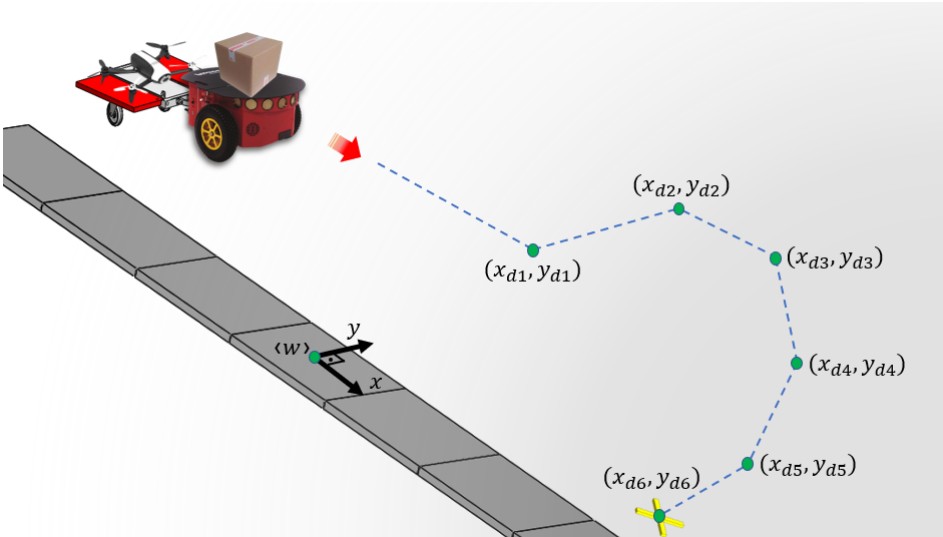

**Figure 5.** Sequence of way-points.

## 4. Robots Modeling and Control

It is worth emphasizing that the focus of this work is the implementation of the controllers to guide a formation of ground and aerial robots cooperating in a load transportation task. Despite its relevance, the design and stability proofing of the controllers are beyond the scope of this paper. We leave for interested readers the details presented in our previous works [29,30]. Nevertheless, to facilitate the understanding of this manuscript and, consequently, its implementation, this section brings the model of the vehicles and the controllers used to guide them.

### 4.1. The Control Strategy for the UGVs

Our UGVs are a unicycle-type mobile robot, whose kinematic model is given by

$$\begin{bmatrix} \dot{x} \\ \dot{y} \\ \dot{\psi} \end{bmatrix} = \begin{bmatrix} \cos\psi & -a\sin\psi \\ \sin\psi & a\cos\psi \\ 0 & 1 \end{bmatrix} \begin{bmatrix} u \\ \omega \end{bmatrix}. \tag{1}$$

Observing Figure 6, $\mathbf{x} = \begin{bmatrix} x & y \end{bmatrix}^\top$ is the robot position, $\psi$ is its heading with respect to $x$-axis of the global frame $\langle W \rangle$, and $a$ is the linear distance between the point of control and the robot axle. Finally, $u$ and $\omega$ are its linear and angular velocities, respectively.

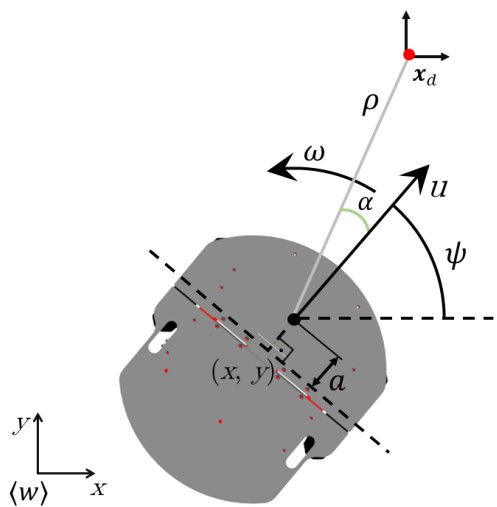

**Figure 6.** The UGV seeking for its destination $\mathbf{x}_d$.

Representing the robot navigation in polar coordinates, we have

$$\rho = \sqrt{(x_d - x)^2 + (y_d - y)^2} \tag{2a}$$

$$\theta = \arctan \frac{y_d - y}{x_d - x} \tag{2b}$$

$$\alpha = \theta - \psi, \tag{2c}$$

where $\rho$ is the robot-destination distance, $\theta$ is the desired orientation, and $\alpha$ is the heading error to the destination.

In this work, the navigation circuit is given by a set of way-points, without time constraints. Consequently, the robot performs a regulation (positioning) task, in which temporal derivatives of the reference positions are equal to zero. Having this in mind, let us take the first time derivative of (2). By replacing (1), we get the kinematic model in polar coordinates

$$\begin{bmatrix} \dot{\rho} \\ \dot{\alpha} \end{bmatrix} = \begin{bmatrix} -\cos \alpha & -a \sin \alpha \\ \dfrac{\sin \alpha}{\rho} & -a \dfrac{\cos \alpha}{\rho} - 1 \end{bmatrix} \begin{bmatrix} u \\ \omega \end{bmatrix}, \quad \text{or in the compact form} \quad \dot{\mathbf{h}} = \mathbf{B}\mathbf{u}, \tag{3}$$

with $\dot{\theta} = \dot{\alpha} + \omega$.

Assuming that the sequence of way-points favors task-oriented arrival, let us design a controller with no final orientation. In this case, theta is a free variable, with the constraint of being finally bounded. Thus, based on Lyapunov theory, let us take the following radially unlimited candidate function

$$V(\mathbf{h}) = \frac{1}{2}\mathbf{h}^\top \mathbf{h} > 0. \tag{4}$$

By applying the feedback linearization technique, we can propose the control signal

$$\mathbf{u} = -\mathbf{B}^{-1}\mathbf{G}\tanh \mathbf{h}, \tag{5}$$

with $\mathbf{G}$ being a positive defined gain matrix.

After applying its first time derivative in (4), and replacing (3) and (5), we get

$$\dot{V}(\mathbf{h}) = \mathbf{h}^\mathsf{T}\dot{\mathbf{h}} = \mathbf{h}^\mathsf{T}\mathbf{B}\mathbf{u} = -\mathbf{h}^\mathsf{T}\mathbf{G}\tanh\mathbf{h} < 0. \tag{6}$$

Therefore, it is demonstrated that the global asymptotic convergence of $\mathbf{h} \to 0$, for $t \to \infty$, which means that the robot will reach its goal in the absence of obstacles. In addition, as a consequence, $\mathbf{u} \to 0$, also for $t \to \infty$.

### 4.2. The Control Strategy for the UAV

Our UAV is the quadrotor Bebop 2, Parrot Inc., in X-configuration, whose body and global frame are illustrated in Figure 7. Its translational coordinates are defined as $\mathbf{x} = \begin{bmatrix} x & y & z \end{bmatrix}^\top$ and its attitude is described by the vector $\boldsymbol{\eta} = \begin{bmatrix} \phi & \theta & \psi \end{bmatrix}^\top$, which contains the Tait–Bryan angles of rotation, pitch and yaw, both related to the global frame $\langle w \rangle$.

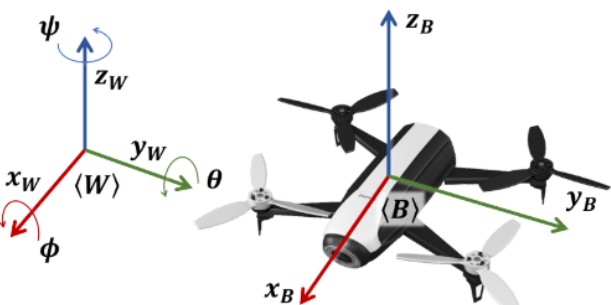

**Figure 7.** Reference system for Parrot Bebop 2 UAV, according to Tait–Bryan angles.

Because Bebop 2 has an autopilot, we can assume that its in-flight stabilization is the responsibility of the embedded firmware. Consequently, for tasks performed in near-hovering operations, the UAV response can be modeled by a more simplistic set of equations, useful for the proposal of flight controllers. Given this fact, its simplified dynamic model in the global frame can be written as

$$\begin{bmatrix} \ddot{x} \\ \ddot{y} \\ \ddot{z} \\ \ddot{\psi} \end{bmatrix} = \begin{bmatrix} k_1\cos\psi & -k_3\sin\psi & 0 & 0 \\ k_1\sin\psi & k_3\cos\psi & 0 & 0 \\ 0 & 0 & k_5 & 0 \\ 0 & 0 & 0 & k_7 \end{bmatrix} \begin{bmatrix} u_{v_x} \\ u_{v_y} \\ u_{\dot{z}} \\ u_{\dot{\psi}} \end{bmatrix} - \begin{bmatrix} k_2\cos\psi & -k_4\sin\psi & 0 & 0 \\ k_2\sin\psi & k_4\cos\psi & 0 & 0 \\ 0 & 0 & k_6 & 0 \\ 0 & 0 & 0 & k_8 \end{bmatrix} \begin{bmatrix} v_x \\ v_y \\ \dot{z} \\ \dot{\psi} \end{bmatrix}, \tag{7}$$

or in the compact formulation as $\ddot{\mathbf{q}} = \mathbf{F}_1\mathbf{u} - \mathbf{F}_2\mathbf{v}$.

As stated in [30], such a model is obviously not a complete description of rotorcraft dynamics. However, this formulation accurately describes the influence of high-level control signals on the vehicle's maneuvers.

From (7), $v_x$ and $v_y$ are the linear velocity along $x-$ and $y-$axis of the UAV body frame, whereas $\dot{z}$ and $\dot{\psi}$ are the linear and angular velocities along and around the $z$-axis, respectively, in the global frame. For experimental purposes, the Bebop's firmware provides the processed data readout of these variables.

Finally, $\mathbf{u} \in [-1, 1]$ is the normalized control signals. In summary, $u_{v_x}$ and $u_{v_y}$ represent pitch and roll commands, which indirectly causes a linear velocity and a displacement along the $x_B$- and $y_B$-axis, respectively. In addition, $u_{\dot{z}}$ and $u_{\dot{\psi}}$ represent the inputs associated with $\dot{z}$ and $\dot{\psi}$, respectively.

In order to perform a positioning task, let us propose the control law

$$\mathbf{u} = \mathbf{F}_1^{-1}(\boldsymbol{\eta} + \mathbf{F}_2\mathbf{v}), \quad \text{with} \quad \boldsymbol{\eta} = \ddot{\mathbf{q}}_d + \mathbf{K}_d\dot{\tilde{\mathbf{q}}} + \mathbf{K}_p\tilde{\mathbf{q}}, \tag{8}$$

where $\tilde{\mathbf{q}} = \mathbf{q}_d - \mathbf{q}$ is the pose error, and $\mathbf{K}_d$ and $\mathbf{K}_p$ are positive defined gain matrices. Notice that this controller design follows the inverse dynamic formulation.

In order to demonstrate the system stability in the closed loop, let us take the following radially unlimited Lyapunov candidate function

$$V(\tilde{\mathbf{q}}, \dot{\tilde{\mathbf{q}}}) = \frac{1}{2}\tilde{\mathbf{q}}^\mathsf{T}\mathbf{K}_p\tilde{\mathbf{q}} + \frac{1}{2}\dot{\tilde{\mathbf{q}}}^\mathsf{T}\dot{\tilde{\mathbf{q}}} > 0. \tag{9}$$

By applying the first time derivative, and then replacing (7) and (8), we get

$$\begin{aligned}
\dot{V}(\tilde{\mathbf{q}}, \dot{\tilde{\mathbf{q}}}) &= \tilde{\mathbf{q}}^\mathsf{T}\mathbf{K}_p\dot{\tilde{\mathbf{q}}} + \dot{\tilde{\mathbf{q}}}^\mathsf{T}(\mathbf{F}_1\mathbf{u} - \mathbf{F}_2\mathbf{v}) \\
&= \tilde{\mathbf{q}}^\mathsf{T}\mathbf{K}_p\dot{\tilde{\mathbf{q}}} + \dot{\tilde{\mathbf{q}}}^\mathsf{T}(-\mathbf{K}_d\dot{\tilde{\mathbf{q}}} - \mathbf{K}_p\tilde{\mathbf{q}}) \\
&= -\dot{\tilde{\mathbf{q}}}^\mathsf{T}\mathbf{K}_d\dot{\tilde{\mathbf{q}}} \le 0.
\end{aligned} \tag{10}$$

From the Lyapunov theory, we can conclude that $\dot{\tilde{\mathbf{q}}} \to 0$ for $t \to \infty$. Furthermore, $\dot{\tilde{\mathbf{q}}}$ and $\tilde{\mathbf{q}}$ are ultimately bounded. Finally, after applying the theorem of La Salle, we can also conclude that the unique invariant set in the closed-loop system is the equilibrium $[\tilde{\mathbf{q}} \ \dot{\tilde{\mathbf{q}}}]^\mathsf{T} = [\mathbf{0} \ \mathbf{0}]^\mathsf{T}$, which means also $\tilde{\mathbf{q}} \to 0$ for $t \to \infty$. Therefore, the regulation (or tracking) system is asymptotically stable.

## 5. Results and Discussion

### 5.1. Experimental Setup

This section describes the strategy used to evaluate the UAV-UGV cooperation in face of ground obstacles for the accomplishment of a load transportation task our configuration setup is show in Figure 8. As can be seen in the figure, the UGVs adopted are the Pioneer 3-DX nonholonomic unicycle mobile platform, and the UAV is the Bebop 2 quadrotor. The posture of the vehicles is tracked by the OptiTrack motion tracking system, consisting of 15 cameras, which are placed around the experimental arena. The data captured by using mocap technology is then used as a constraint to reduce the ambiguity of the real-time inverse kinematic process by capturing the posture of all elements that make up the experiment and thus updating the control signals. To communicate with the robots, the robot operating system (ROS) is used. As for the high-level control code, it was executed by using MATLAB©. Communication between ROS and MATLAB is done through a proprietary MATLAB library, which emulates the necessary ROS nodes and topics. The positions of the load, the electromagnet, and the trailer are also captured by the OptiTrack system.

Once this work aims to perform a load transportation task, the electromagnet actuator is one of the main components of our system. However, the device should not have a permanent magnet, because the proposal considers capture and release actions only at opportune moments. Furthermore, to reduce the weight of the device, which must be on board the UAV, we have chosen to trigger it remotely. Thus, the capture and release action is decided externally and transmitted via wireless to the actuator.

The electromagnet has a capacity of up to 2.5 kg and runs on DC power through a two-cell, 7.4-V, 300-mA lithium polymer battery. The receiver board that controls the operation of the electromagnet is controlled by an Arduino Nano microcontroller. Furthermore, such a board allocates all the necessary components on a compact-sized board, so as not to interfere with the controllability of the UAV during flight. Table 1 lists the components used to make the transmitter and receiver, shown in Figure 9. Note that there is a yellow LED in Figure 9a. If it is on, the electromagnet is on; otherwise, it is off.

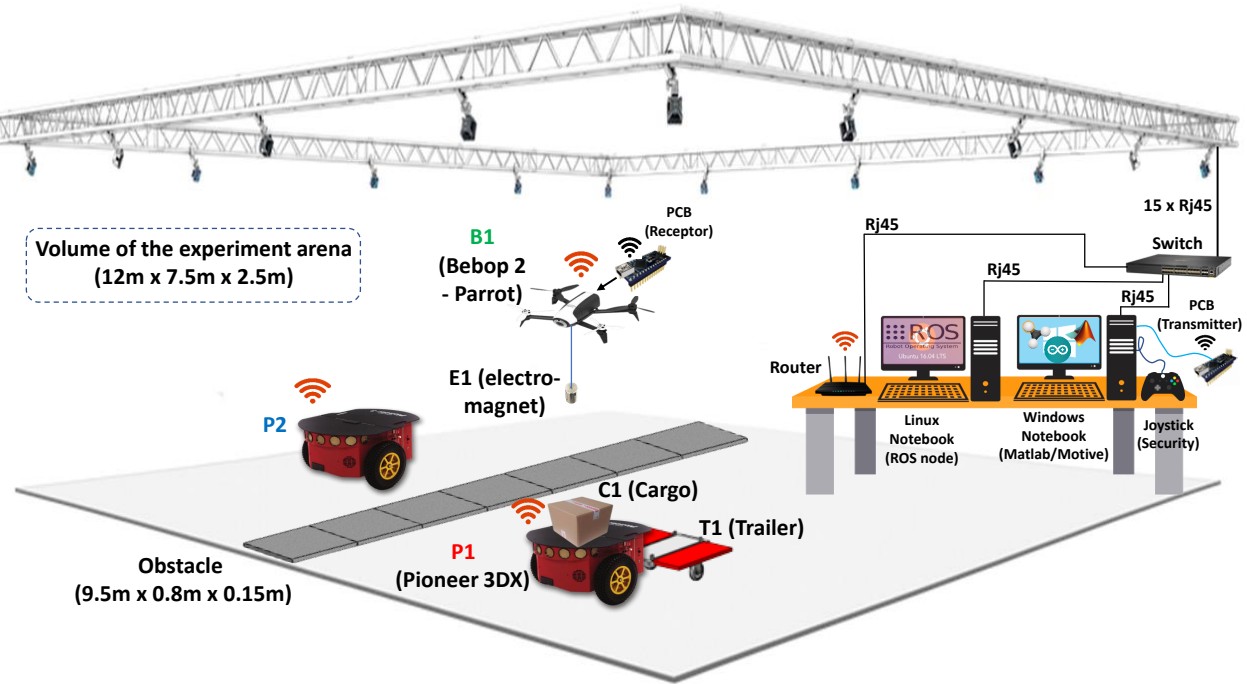

**Figure 8.** The experimental setup.

**Table 1.** Table of components used in transmitter and receptor.

| Component | Quantity | Present in Transmitter (T) or Receptor (R) |
|---|---|---|
| Arduino Nano | 2 | T and R |
| Battery | 1 | R |
| TBJ 538 | 1 | R |
| Electromagnet | 1 | R |
| LED | 1 | T |
| Push Button | 1 | T |
| Resistors 220 Ω | 2 | T and R |
| Switch | 1 | R |
| Radio NRF24L01 | 2 | T and R |
| Antenna for WiFi Module | 1 | R |
| 2.4 GHz Wireless Transceiver Module—NRF24L01 | 2 | T and R |

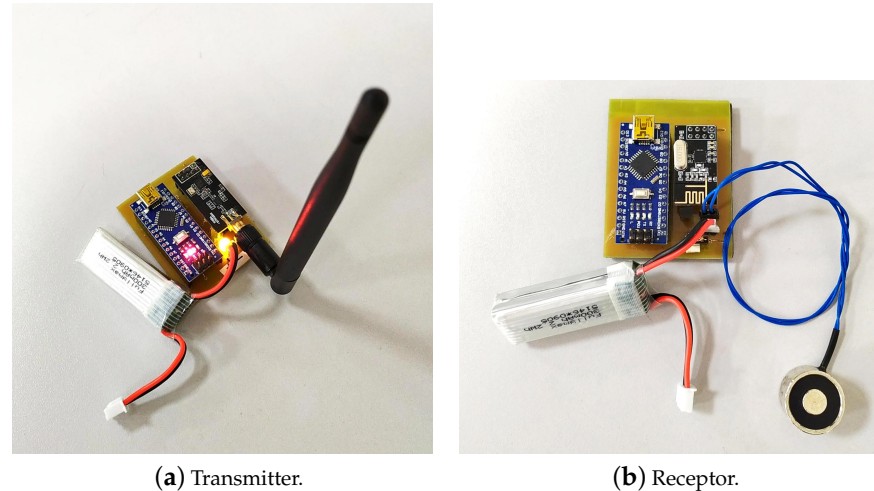

(**a**) Transmitter.          (**b**) Receptor.

**Figure 9.** Printed circuit boards (PCBs) for electromagnet actuation.

Importantly, one of the biggest challenges faced with suspended cable transportation is the effect of the load swinging after being captured by the electromagnet device. In other words, the actuator must be able to support the weight of the load even when the swing effect is present as a consequence of the UAV's displacement maneuvers. In the case of this work, cautious movements, with small angles of tilt and rotation of the UAV, helped mitigate the oscillations.

In terms of the load, it has a cubic shape of 10 cm side and weight of 72 g. Figure 10 shows two configurations: empty and full cargo. In the last case, a set of five styrofoam block increases the cargo weight by 25 g. The metal plate over the cargo prototype makes possible its interaction with the magnetic actuator.

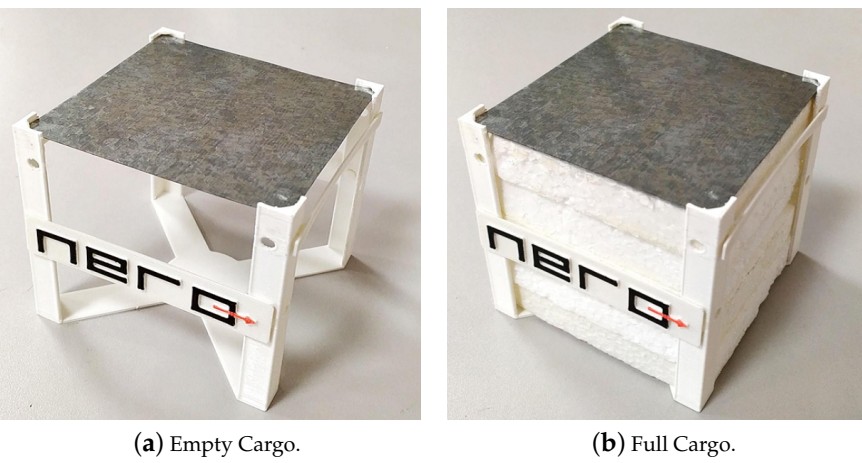

(**a**) Empty Cargo.　　　　　　　　　　　　(**b**) Full Cargo.

**Figure 10.** Cargo transported in experiments.

*5.2. Cooperative Task between UAV and UGVs*

The following experiments examine the influence of the perturbation generated by the mass change during the holding and release action, and the ability of the system to handle it. For a better explanation, we divide our discussions into two subsections, wherein transport with loads of different weights are evaluated. In both cases, a heterogeneous formation with one UAV and two UGVs makes up the experiments.

5.2.1. Experiment #1

In the first experiment, the weight of the actuator and the payload is 90 g and 72 g, respectively, which corresponds to about 32% of the weight of the drone. The video presenting the entire experiment can be found at https://youtu.be/0UIOjn5_a5s (accessed on 6 December 2022). Table 2 presents in chronological sequence in which the robots perform each action cooperatively.

Experiment #1 is divided temporally into five regions: transportation, setup, lifting, returning, and transportation again. These regions make up the complete realization of the experiment. In addition, Figure 11 highlights the actions that B1 performs at each region transition: take off, docked, released and landing, respectively.

At the beginning of the experiment, only P1 starts its motion by carrying C1 and B1 through the T1 trailer. This step comprises the transportation region and occurs in the interval from 0 to 42 s. After 20 s, P1 reaches a previously defined desired point in an orientation perpendicular to the obstacle. At this instant, P2 comes from rest and starts moving to reach a position in front of P1 and also perpendicular to the obstacle.

Starting at 42 s, the setup region is initiated and occurs in the interval from 42 to 55 s. During this interval, B1 takes off, aiming to position itself under C1 in order to take it. The change in altitude of B1 in Figure 11 confirms such an action. Note also that before taking off, B1 is at a height close to 30 cm; after all this is the height of T1 added to the height of B1's center of gravity relative to T1.

In the interval from 52 to 55 s, it can be seen that E1 and C1 have the same vertical coordinate, i.e., the actuator is in contact with the load. This situation is maintained until the load is released.

**Table 2.** Sequence of events of the experiment 1.

| Seq | Time (mm:ss) | Region | Action | Snapshot (mm:ss) |
|---|---|---|---|---|
| 1 | 00:01 | Transportation | P1 carries (C1 + B1) to the combined point | 01:25 |
| 2 | 00:22 | Transportation | P1 arrives at the agreed point and triggers P2 which starts moving | 01:47 |
| 3 | 00:42 | Docked \| Lifting | P2 reaches the combined point and triggers B1 to take off | 02:07 |
| 4 | 00:55 | Lifting | B1 picks up C1 using the electromagnet E1 and go to P2 | 02:20 |
| 5 | 01:08 | Lifting \| Returning | B1 releases C1 the under P2 and returns to the other side of the platform | 02:33 |
| 6 | 01:18 | Returning \| Transportation | B1 positions himself above the trailer and lands | 02:43 |
| 7 | 01:40 | Transportation | The formation returns to their respective address and the task is successfully completed | 02:50 |

Upon entering the lifting region, which occurs after 55 s, B1 grasps C1 and makes its transposition between P1 and P2, over the obstacle. It is worth remembering that P1 and P2 wait at rest until the transposition of the load is successfully completed. Notice in Figure 11 that the altitude of E1 and C1 are identical during this stage, which confirms that the payload is being lifted by the UAV. Note also that the distance between B1 and C1 represents the length of the suspended cable.

At the end of the lifting stage, note that C1 returns to the same altitude value that it had before being lifted from P1. This demonstrates that the load has been delivered to P2, and is confirmed by the change in the side elevation $y$, as seen in Figure 11.

At 67 s, B1 releases C1 over P2 and the returning region is initiated. After the delivery, B1 returns to land on T1. Finally, in the last region called transportation, which starts at 78 s, the entire formation moves to its destination point, successfully completing the entire transport task.

Note that the altitude variation observed in Figure 11, around 80 s, indicates B1's attempts to land on T1 and to have succeeded on the second attempt. It generally depends on how aligned B1 is with respect to T1.

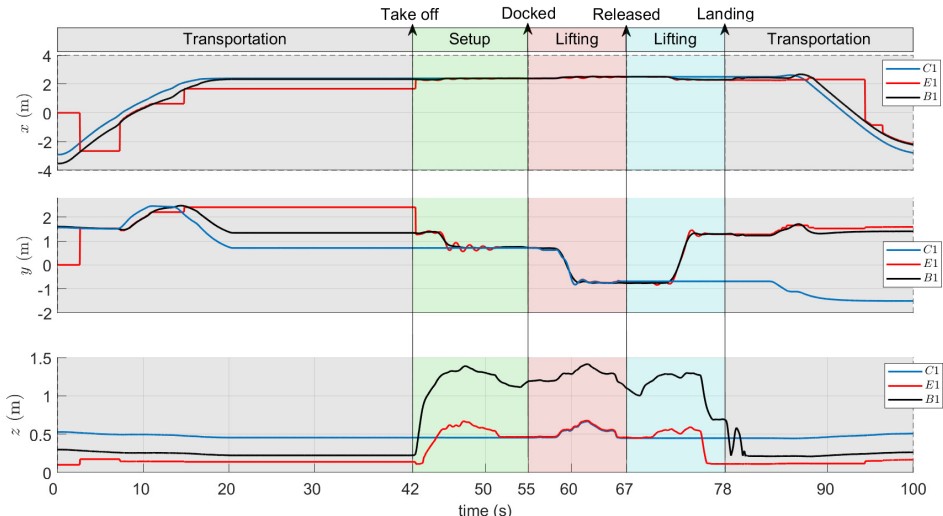

**Figure 11.** Results of Experiment #1. Position time evolution of the cargo (C1), the electromagnet effector (E1), and the Bebop UAV (B1).

### 5.2.2. Experiment #2

In the second experiment, the weight of the load is increased to 97 g. Consequently, the total weight to be lifted corresponds to about 37% of the weight of the UAV. The additional 25 g are used to evaluate the controller response under the same conditions of Experiment #1. The whole evaluation can be seen at https://youtu.be/0UIOjn5_a5s (accessed on 6 December 2022).

Table 3 presents the instants of the events of each action referring to Experiment #2. All the analysis and discussion of Experiment #1 also apply to Experiment #2. Notice that this time the landing of B1 on T1, after the delivery of C1, occurred on the first attempt. This can be seen in the altitude variable of B1 around 80 s in Figure 12.

Comparatively, during the transport stage, the load showed greater longitudinal and lateral oscillations in Experiment #2 than in #1. This is a consequence of the additional mass of the load causing the oscillation amplitude to increase. However, the controller proved to be able to handle this disturbance.

**Table 3.** Sequence of events of the experiment 2.

| Seq | Time (mm:ss) | Region | Action | Snapshot (mm:ss) |
|---|---|---|---|---|
| 1 | 00:01 | Transportation | P1 carries (C1 + B1) to the combined point | 03:15 |
| 2 | 00:22 | Transportation | P1 arrives at the agreed point and triggers P2 which starts moving | 03:33 |
| 3 | 00:42 | Docked \| Lifting | P2 reaches the combined point and triggers B1 to take off | 03:53 |
| 4 | 00:55 | Lifting | B1 picks up C1 using the electromagnet E1 and go to P2 | 04:07 |
| 5 | 01:08 | Lifting \| Returning | B1 releases C1 the under P2 and returns to the other side of the platform | 04:21 |
| 6 | 01:18 | Returning \| Transportation | B1 positions himself above the trailer and lands | 04:30 |
| 7 | 01:40 | Transportation | The formation returns to their respective address and the task is successfully completed | 04:34 |

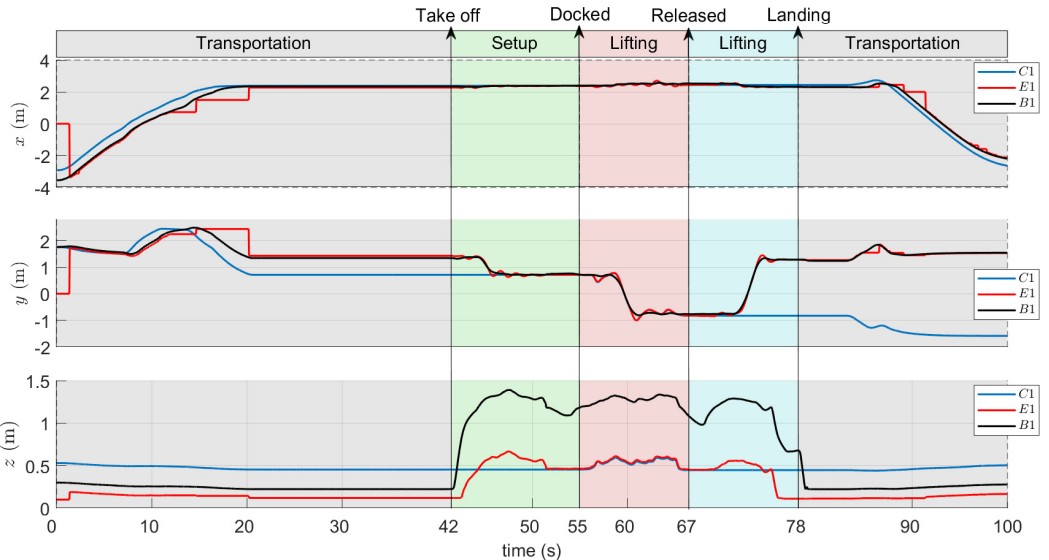

**Figure 12.** Results of Experiment #2. Position time evolution of the cargo (C1), the electromagnet effector (E1), and the Bebop UAV (B1).

Both experiments showed that it is possible to successfully coordinate a heterogeneous squadron to together perform a cargo transport mission, with the possibility of the UAV taking off and landing on a trailer connected to one of the UGVs. Furthermore, this work showed the application of an electromagnetic actuator connected to an overhead cable as an option to grab and release the cargo at the opportune moments of the transport

mission. Finally, being an applied science project, the results demonstrated the stability of the controllers proposed in previous works, as well as their tolerance to the rejection of disturbances caused by load transportation.

## 6. Concluding Remarks

This paper has presented a practical application of a heterogeneous formation of robots in performing a task of lifting, transporting, and delivering cargo by suspended cables. The problem addressed was the impossibility of transporting a load by using only ground robots, due to the presence of an obstacle that prevented its transposition. Therefore, a UAV capable of flying was in charge of transporting the load over the obstacle and delivering it to the destination robot.

The control strategies proposed in our previous work were used here to guide the vehicles. For the ground robots, the controller guided the vehicles to the meeting and destination points as planned. Notably, the way-points contributed to the robot-trailer set during its backward maneuvers. On the other hand, for the aerial robot, the controller proved to be capable of controlling the UAV, even in the presence of a load beyond its own weight. During navigation with lighter or heavier loads, small oscillations were observed in the lateral and longitudinal movements, but nothing that would compromise the stability of the set or even the accomplishment of the mission.

Finally, the electromagnetic effector developed proved to be a simple, economical, and efficient solution to perform cargo transport tasks. Although its load capacity is 2.5 kg, the lifting or dragging of such a mass could not be tested due to the load capacity of the UAV used. Thus, transporting heavier loads in more varied ways remains as an idea for future work.

**Author Contributions:** Conceptualization, C.O.B. and L.A.F.-J.; methodology, C.O.B.; validation, C.O.B., L.A.F.-J. and A.S.B.; resources, A.S.B., A.P.S., M.S.-F. and D.C.G.; data curation, C.O.B. and L.A.F.-J.; writing original draft preparation, C.O.B. and L.A.F.-J.; writing, C.O.B. and L.A.F.-J., review and editing, A.S.B., D.K.D.V., D.C.G., A.P.S. and M.S.-F.; supervision, A.S.B.; project administration, A.S.B.; funding acquisition, A.S.B., A.P.S. and M.S.-F. All authors have read and agreed to the published version of the manuscript.

**Funding:** This work was financially supported by FAPEMIG—Fundação de Amparo à Pesquisa do Estado de Minas Gerais (Grant Number APQ-02573-21).

**Institutional Review Board Statement:** Not applicable.

**Informed Consent Statement:** Not applicable.

**Data Availability Statement:** Not applicable.

**Acknowledgments:** Barcelos and Fagundes-Junior thank CAPES—Coordenação de Aperfeiçoamento de Pessoal de Nível Superior and FAPEMIG—Fundação de Amparo à Pesquisa do Estado de Minas Gerais, respectively, for their scholarships.

**Conflicts of Interest:** The authors declare no conflict of interest.

## Abbreviations

The following abbreviations are used in this manuscript:

| | |
|---|---|
| UAV | Unmanned Aerial Vehicle |
| UAV | Unmanned Ground Vehicle |
| B1 | UAV 1 (Bebop) |
| P1 | Pioneer 1 |
| P2 | Pioneer 2 |
| C1 | Cargo |
| E1 | Electromagnet |

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
