# Peer review of "Robot Formation Performing a Collaborative Load Transport and Delivery Task by Using Lifting Electromagnets"

_applsci, doi:10.3390/app13020822_

Round 1

Reviewer 1 Report

The main concern is that there needs to be a better description of the control strategy that is being implemented. For this type of article, I don't think that it is enough to cite the author's other article where it is explained in depth. At least a summarized version should be included in this article and then referencing the other article would be OK, as it would be for additional details.

On page 2, I suspect the word "monabrability"  is a typo for "manoeuvrability".

On page 5, the sentences "The motion capture system is used to measure the vehicle’s positions along its navigation. OptiTrack, configured with fifteen cameras scattered around the room." are not easy to follow. They should be reestructured.

Author Response

Please understand (R1C1) as the answer for the Comment #1 of Reviewer #1. 

 (R1C1) 

We greatly appreciate your comment, since we understood that the simple fact of citing the theoretical reference was already sufficient for the understanding of the rest of the article. For understanding and acknowledging the need for better detailing, the new section was created with the description of the models of the aerial and ground mobile robots, the design and proposal of the controllers, and their stability demonstration. Thus, we hope that the new Section IV has addressed your valuable suggestion.

(R1C2) 

Checked and corrected.

(R1C3)

Thanks for your comment. We reviewed the sentence and restructured it as:

"The posture of the vehicles is tracked by the Optitrack motion tracking system, consisting of 15 cameras, which are placed around the experimental arena."

Reviewer 2 Report

The work is interesting and shows the cooperation of robots. The experiments were presented on YouTube, which additionally significantly enriched and showed the content of the article. The solution can be very helpful as a support for rescue systems. It is worth presenting this solution. The article is written quite correctly.

Author Response

Thank you very much for considering our work interesting for Applied Science readers, and especially for the robotics science community. Thank you for reviewing the article and mainly for the positive feedback.

Reviewer 3 Report

Authors provided a practical validation of a UAV-UGV cooperation in performing a load lifting, transportation and delivery task. After reading this article, we found that the author did not discuss the scientific problems in this application.Not to mention the innovation of this article. Even there is no discussion for controller design and stability, which should be found in another paper of the author. Therefore, I do not understand the value and significance of this paper.

Author Response

First of all, we would like to thank you for reviewing the article and for pointing out what we should insert to improve our manuscript. In the submission process, we focused on developing a paper with a practical application related to the challenges of load transportation with a heterogeneous formation, acting cooperatively. After this round of review, we realized the need to detail the proposal of the controllers, as well as the stability demonstration for each of them. Therefore, we hope to have taken into account your valuable suggestion, with the new Section IV of the new version of our manuscript.

Round 2

Reviewer 3 Report

although authors provided some  controllers design, as well as the stability demonstration for each of them to Enhance the attribute of scientific journal papers rather than popular science articles. However the Scientific innovation is still not enough.  Many logistics companies use unmanned aerial vehicles (i.e., heterogeneous robots) in formation to perform collaborative load transfer and delivery tasks using lifting electromagnets. Therefore, it is not a very novel application. 

But the author completed the experimental verification and showed it on Youtube. Maybe it has some value.